abiotic and biotic stresses; DNA methylation; epigenetics; epialleles; intergenerational response; stress tolerance; transgenerational response.

**Corresponding author:**
Igor Kovalchuk;
Email: Igor.kovalchuk@uleth.ca

# Heritable responses to stress in plants

## Igor Kovalchuk

Department of Biological Sciences, University of Lethbridge, Lethbridge, AB, Canada

### Abstract

Most plants are adapted to their environments through generations of exposure to all elements. The adaptation process involves the best possible response to fluctuations in the environment based on the genetic and epigenetic make-up of the organism. Many plant species have the capacity to acclimate or adapt to certain stresses, allowing them to respond more efficiently, with fewer resources diverted from growth and development. However, plants can also acquire protection against stress across generations. Such a response is known as an intergenerational response to stress; typically, plants lose most of the tolerance in the subsequent generation when propagated without stress. Occasionally, the protection lasts for more than one generation after stress exposure and such a response is called transgenerational. In this review, we will summarize what is known about inter- and transgenerational responses to stress, focus on phenotypic and epigenetic events, their mechanisms and ecological and evolutionary meaning.

## 1. Introduction

Through millions of years of evolution, organisms developed mechanisms of stress avoidance, resistance and tolerance and became well-adapted to their environment. These responses are encoded by the genetic make-up of the organism and are fine-tuned by epigenetic regulations. To be able to respond to the environment in a manner similar to their ancestral generations, the progeny requires faithful replication of their genetic material and epigenetic marks. This is critically important for the survival of an organism in a stable environment. In contrast, survival under stressful conditions requires drastic measures that are implemented quickly. In such a situation, due to the rare nature of mutations, genetic mechanisms may not be able to provide swift and efficient responses for the survival of future generations. In contrast, the regulation at the epigenetic level represents a more versatile and flexible mechanism controlling gene expression and inheritance of old traits and the appearance of new traits (Chang et al., 2020). Epigenetic mechanisms are frequently reversible because they do not represent permanent chemical changes (Tao et al., 2017). Moreover, the rate and spectrum of epigenetic changes by far exceed those of genetic changes, allowing better phenotypic plasticity and faster adaptation (van der Graaf et al., 2015).

When environmental conditions become substantially different from normal, the plant employs various mechanisms, including epigenetics, to pass the memory of responses to encountered stresses to the progeny (Nguyen et al., 2022). This information may be in the form of differentially accumulated metabolites, including primary and secondary metabolites (proteins, fatty acids, messenger ribonucleic acid (mRNA), non-coding RNA (ncRNA), etc.) or chromatin modifications in the form of deoxyribonucleic acid (DNA) methylation or histone modifications. The most well-known examples of such response to stress are known as adaptation and acclimation (Ding et al., 2020). Changes observed in the progeny are often referred to as intergenerational (Lamke & Baurle, 2017) or transgenerational stress response, but they also have several other names, including intergenerational inheritance, intergenerational resilience, plasticity, priming or tolerance. For the sake of this review, we will refer to the changes observed in the immediate progeny of exposed plants as intergenerational changes (IGCs) (Verhoeven et al., 2018). In contrast, when the changes persist to the 'grand progeny', without stress, we will refer to them as transgenerational changes (TGCs). Furthermore, in this review, we will not cover such classical transgenerational events as paramutations (Heard & Martienssen, 2014).

Many parameters likely regulate the ability to establish IGC or TGC, including the species analysed, genetic and epigenetic composition, type of stress, severity of stress, length of the exposure and time during the development when plants were exposed. Also, IGC and TGC manifest themselves as changes in transcriptome, in DNA methylation pattern, in plant physiology and in plant response to stress. We will discuss these points in detail in this review and introduce potential mechanisms of the establishment of heritable memory of stress exposure.

## 2. IGCs and TGCs

### 2.1. Types of IGCs or TGCs

IGC and TGC may include alterations at many levels: DNA methylation and histone modifications, changes in transcriptome, including mRNA and ncRNA transcripts, and changes in metabolome and proteome (reviewed in Herman & Sultan, 2011; Kinoshita & Seki, 2014; Liu et al., 2019). These changes, when occurring in response to stress, typically lead to higher tolerance to the same or similar stresses but may also result in increased, and sometimes decreased, tolerance to other stresses, for example, higher tolerance to heat stress, but lower tolerance to pathogens. Such changes often disappear in consecutive generations, and most likely, they occur due to differential seed viability or quality caused by the accumulation of metabolites or nutrients, such as starch, hormones, such as abscisic acid and other primary and secondary metabolites that give a certain advantage to plants grown under specific environmental conditions (Donohue, 2009). Occasionally, especially in cases when the stressor persists for longer, TGCs persist, in the form of epialleles. The only well-documented types of so-called natural epialleles are those due to changes in DNA methylation (van der Graaf et al., 2015). It is important to distinguish such naturally occurring epialleles from IGC and TGC events triggered experimentally. While IGC and TGC observed experimentally cover all the above-mentioned changes, the naturally occurring epialleles only retain changes in methylation patterns. It is possible that some naturally occurring epialleles are the result of spontaneous events, possibly mutations in the components of the epigenetic machinery, leading to heritable epigenetic change. We hypothesize, however, that most naturally occurring epialleles are the consequences of changes in the environment, 'forcing' an entire population or a sub-population of plants to acquire an epiallele. In this respect, the TGCs we observe when we conduct experiments are the initial steps towards the formation of epialleles.

### 2.2. Naturally occurring epialleles as evidence of TGCs

TGCs may be heritable and even persist for many generations, forming epialleles (Quadrana & Colot, 2016; Tonosaki et al., 2022; van der Graaf et al., 2015). Such epialleles typically consist of differentially methylated loci, where cytosines at various positions are hyper- or hypomethylated as compared to the parental alleles. Many known epialleles are believed to have occurred naturally (Table 1). In *Linaria vulgaris*, hypermethylation of linaria cycloidea-like gene (*Lcyc*), the gene responsible for flower symmetry, results in a stable phenotype, which reverts occasionally upon a loss of hypermethylation (Cubas et al., 1999). Imprinting of the FLOWERING WAGENINGEN (*FWA*) gene in Arabidopsis results in female-specific expression, causing a stable late flowering phenotype (Fujimoto et al., 2008). Variations in methylation of a retrotransposon, named 'NMR19' (naturally occurring DNA methylation variation region 19), represent epialleles that control leaf senescence by regulating the expression of *PHEOPHYTIN PHEOPHORBIDE HYDROLASE* (*PPH*) in Arabidopsis (He et al., 2018). The NMR19-4 epiallele is heritable and correlates with local climates (He et al., 2018). In rice, heritable hypomethylation in the *FERTILIZATION-INDEPENDENT ENDOSPERM1* (*FIE1*) gene results in the dwarf phenotype (Zhang et al., 2012). *FIE1* encodes a component of the polycomb repressive complex 2 involved in H3K27me3-mediated gene repression; this naturally occurring gain-of-function hypomethylation results in the change in histone modifications of hundreds of genes (Zhang et al., 2012). Another case of heritable DNA hypermethylation involves the *colourless non-ripening* (*Cnr*) gene, responsible for the fruit ripening and colouring in tomatoes (Manning et al., 2006). In the perennial herb *Helleborus foetidus*, many heritable size- and fecundity-related traits are controlled by DNA methylation (Alonso et al., 2014). Yet, another example of heritable epigenetic changes includes the *de novo*-originated gene *qua-quine starch (QQS)* in *Arabidopsis thaliana*; Silveira et al. (2013) found substantial variations in DNA methylation in natural accessions of Arabidopsis, with many hypomethylated states inherited for up to eight generations (Silveira et al., 2013).

### 2.3. IGCs or TGCs in the form of changes in the plant stress response

As stated above, IGC and TGC manifest themselves in various forms, with the most common being changes in phenotype, stress tolerance and epigenetic modifications. The most desired effect of IGC or TGC is an increased stress tolerance that does not affect the plant performance under normal conditions.

**Table 1.** Naturally occurring epialleles.

| Plant | Targeted gene | Type of modification | Phenotype | References |
|---|---|---|---|---|
| *Linaria vulgaris* (yellow toadflax) | *Lcyc* | DNA hypomethylation | Change in flower symmetry | Cubas et al. (1999) |
| *Arabidopsis thaliana* (Arabidopsis) | *FWA* | DNA methylation; imprinting | Late flowering phenotype | Fujimoto et al. (2008) |
| Arabidopsis | *PPH* | DNA hypomethylation in the NMR19 transposon | Leaf senescence | He et al. (2018) |
| *Oryza sativa* (rice) | *FIE1* | DNA hypomethylation leading to reduced H3K27me3 and H3K9me2 levels | Dwarf phenotype | Zhang et al., (2012) |
| *Solanum lycopersicum* (tomato) | *Cnr* | DNA hypermethylation | Fruit ripening | Manning et al. (2006) |
| *Helleborus foetidus* (stinking hellebore) | Multiple loci | DNA methylation | Size- and fecundity-related traits | Alonso et al. (2014) |

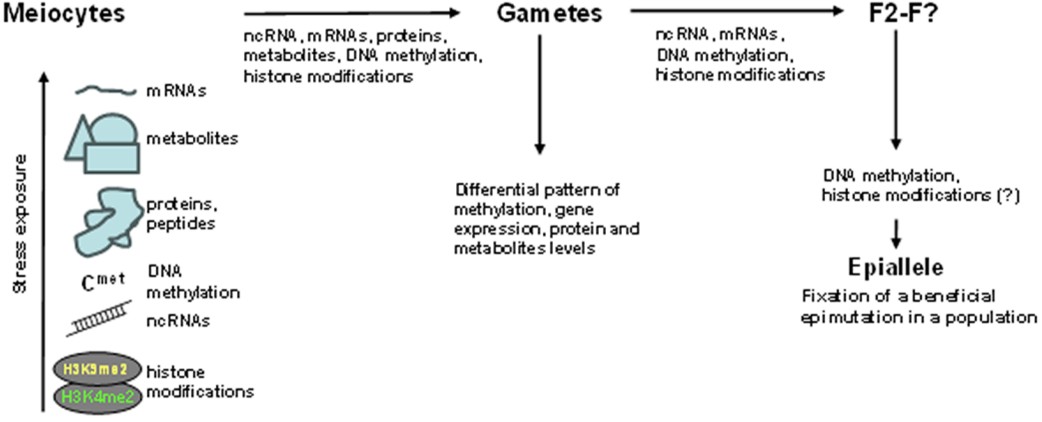

**Figure 1.** Potential mechanism of establishment of transgenerational effects and development of new epialleles. In the proposed scenario, stress generates mobile response molecules, likely in the form of small interfering RNAs (siRNAs) or other types of ncRNAs, but could also include differential levels of proteins, metabolites and various histone modifications, which reach meiocytes and alter DNA methylation and gene expression patterns. Developing meiocytes may retain certain signals and pass new epigenetic patterns into gametes. It is possible that some of the differentially expressed ncRNAs, as well as mRNAs, especially from female gametes, are preserved and influence the developing progeny. The persistence of stress may further reinforce these signalling molecules, leading to the development of stable changes in DNA methylation that do not revert even when stress is absent. Such changes in DNA methylation and chromatin structure represent epimutations and could lead to the development of epialleles persisting for many generations.

Several studies have found that exposure to elevated $CO_2$ levels has a transgenerational effect on plant biomass (Bezemer et al., 2004; Lau et al., 2008; Li et al., 2017; Lv et al., 2022). The immediate progeny of *Poa pratensis* exposed to high $CO_2$ level exhibited higher biomass and produced more tillers (Bezemer et al., 2004). IGC in response to elevated $CO_2$ and increased N (nitrogen) deposition was observed in *Lupinus perennis*, *Poa pratensis* and *Schizachyrium scoparium* (Lau et al., 2008). In particular, the authors found increased biomass and higher seed weight in the progeny of plants exposed to high $CO_2$ when they were grown in the presence of high $CO_2$ or N; curiously, the progeny of plants exposed to high N did not perform better in response to high N, but did do better when grown on high $CO_2$ as compared to the progeny of plants grown in normal $CO_2$ and N (Lau et al., 2008). Two generations of exposure of wheat (*Triticum aestivum* L.) to elevated $CO_2$ resulted in increased stomatal conductance and drought tolerance (Li et al., 2017). In contrast, Lv et al. (2022) found that five consecutive generations of rice exposure to elevated $CO_2$ levels resulted in a decreased rate of photosynthesis and a negative effect on plant growth (Lv et al., 2022).

Klironomos et al. (2005) studied the effect of 21 generations of a perennial grass *Bromus inermis* exposure to elevated $CO_2$; they analysed the response of mycorrhizal symbiotic system to abrupt (from ambient 350 p.p.m. to 550 p.p.m.) or gradual (10 p.p.m. increase per generation, from 350 p.p.m. to 550 p.p.m.) increase in $CO_2$ concentration (Klironomos et al., 2005). The authors did not find any difference between the generation 21 and generation 1 plants in the biomass or photosynthesis rate, while they found that exposure to an abrupt change in $CO_2$ resulted in a significant decrease in biodiversity as compared to ambient $CO_2$ or a gradual change in progeny (Klironomos et al., 2005).

More recent data demonstrated that exposure of *Arabidopsis thaliana* and *Physcomitrium patens* to high $CO_2$ resulted in accelerated growth rates in the immediate progeny (Panda et al., 2023). The authors showed that this intergenerational effect was dependent on DNA methylation, the function of RNA-dependent DNA

methylation (RdDM) machinery and Chromomethyltransferase 2 (CMT2) and CMT3 DNA methyltransferases (Panda et al., 2023).

The progeny of *Oryza sativa* L. exposed to heavy metals was found to be more tolerant to the same stress (Ou et al., 2012). Increased tolerance to heavy metal stress was also found in the progeny of rice plants exposed to heavy metals; the authors also found changes in the expression of various transporters and these changes were also observed in the second generation, when plants were propagated in normal conditions (Cong et al., 2019).

Arabidopsis plants were exposed to salt for five generations, and the authors found evidence of higher tolerance to salt only starting from the second generation, while no such adaptation was found in the first generation after stress exposure (Wibowo et al., 2016). They also noted that the removal of stress at any generation resulted in the loss of this tolerance in the progeny, indicating a transient nature of this change, or an IGC (Wibowo et al., 2016).

The progeny of oilseed rape exposed to drought showed lower quality of seeds but higher tolerance to drought (Hatzig et al., 2018). Similar results were found for rice; exposure to drought for 11 generations improved drought tolerance and oxidative stress resilience (Zheng et al., 2017). Also, the progeny of *Polygonum persicaria* plants exposed to drought had longer roots and larger biomass (Herman & Sultan, 2011). The authors found the effect of two successive generations of drought stress to be cumulative, resulting in greater provisioning, root growth and survivorship when the progeny was exposed to stress. A positive effect on seedling development was even observed when the progeny of stressed plants was propagated in normal conditions, indicating TGC at least for some traits (Herman et al., 2012).

The immediate progeny of ultraviolet C (UVC)-treated Arabidopsis plants exhibited an increase in the seed size, a decrease in the leaf number and an earlier bolting time (Migicovsky & Kovalchuk, 2014). Similar changes were found in the progeny of heat-stressed plants (Migicovsky et al., 2014). Earlier bolting, larger seeds and changes in the leaf number or size are likely some of the mechanisms of adaptation to UV stress.

Higher tolerance to stress was also observed in the progeny of plants infected with pathogens. Luna et al. (2012) found that the progeny of plants infected with *Pseudomonas syringae* exhibited reduced bacterial colonization when encountering similar infection and higher tolerance to a fungal pathogen *Hyaloperonospora arabidopsidis (*Luna et al., 2012*)*. This IGC became a TGC when plants were propagated for one more generation without stress— a higher pathogen tolerance was observed (Luna et al., 2012). Slaughter et al. (2012) confirmed the finding by Luna et al. (2012) in the establishment of IGC in response to infection with *Pseudomonas syringae* but found that the propagation without stress removed this tolerance; thus, no TGC was established (Slaughter et al., 2012). IGC events in the form of cross-tolerance to infection with various pathogens seem to be common. The progeny of tobacco plants infected with tobacco mosaic virus (TMV) not only exhibited higher tolerance to TMV infection but also to inoculation with the bacterial pathogen *Pseudomonas syringae* and the fungal pathogen *Phytophthora nicotianae* and higher tolerance to the chemical methyl methanesulfonate (MMS) (Kathiria et al., 2010).

Insect grazing also led to heritable events. The progeny of wild radish exposed to herbivores was resistant to herbivory (Agrawal, 2001). Also, yellow monkeyflower plants respond to herbivory with an increased trichome density in the progeny; trichome density positively correlates with tolerance to herbivores (Holeski et al., 2010). Exposure of Arabidopsis and tomato plants to caterpillar herbivory resulted in enhanced resistance to two of three herbivores tested in the progeny (Rasmann et al., 2012). This effect was partially transmitted to the next generation when plants were propagated in normal conditions but was lost when they were propagated to the third generation (Rasmann et al., 2012). Also, exposure of *Solanum carolinense* to caterpillar herbivory led to greater emergence, earlier flowering and larger seed yield in the progeny (Nihranz et al., 2020). Wounding often mimics the attack by insect; the progeny of wounded plants exhibited higher trichome density and herbivore resistance (Colicchio, 2017). Also, chemicals mimicking pathogen attack, such as jasmonic acid (JA), trigger heritable changes; dandelion plants treated with JA showed heritable changes in the transcriptomes and metabolomes; the intergenerational effect of treatment was very substantial—about 40% of changes in transcriptome and 10% of changes in metabolome were heritable (Verhoeven et al., 2018).

## 2.4. Changes in DNA methylation in the progeny of stressed plants

Heritable changes in DNA methylation in response to stress have been observed in many reports. A dose-dependent genome hypermethylation was found in the pine trees grown in the Chernobyl area—the progeny germinated from seeds of trees grown in areas with higher radiation load was more hypermethylated (Kovalchuk et al., 2003). More recently, it was shown that the exposure of Arabidopsis plants for three generations to different levels of radiation also resulted in an increase in DNA methylation, primarily in the CG context; the authors noted that the highest level of radiation was less efficient in the establishment of IGC in DNA methylation (Laanen et al., 2021).

Genome hypermethylation was observed in the progeny of Arabidopsis plants exposed to salt for five generations (Wibowo et al., 2016). They found that these methylation changes occurred primarily in CHG and CHH contexts and these changes correlated well with stress treatment, whereas changes in CG methylation patterns occurred stochastically (Wibowo et al., 2016).

In rice exposed to drought for 11 generations, changes in DNA methylation were not linear, with the largest change observed between generations 10 and 11 (Zheng et al., 2017). They found that hypomethylation occurred primarily at CG and CHG contexts at intergenic regions, while hypermethylation occurred mainly in CHH associated with transposable elements. The recurring methylation changes observed in all generations were predominantly at CHH. Finally, they found DNA methylation changes maintained in the progeny propagated in normal watering condition after 11 generations of draught exposure, a transgenerational event (Zheng et al., 2017).

Changes in CHG methylation were also inherited in rice exposed to heavy metals—hypomethylation of cytosines in the CHG context was found (Ou et al., 2012). Exposure of rice to various heavy metal salts showed a complex pattern of changes in DNA methylation in several transposons in the progeny, and these changes persisted to a second generation when plants were propagated in normal conditions, again, a TGC (Cong et al., 2019).

The role of DNA methylation in the establishment of IGCs in *Polygonum persicaria* plants was also shown in the response to drought; while the progeny of drought-exposed plants showed IGC, treatment with the demethylation agent zebularine removed the adaptive advantage, indicating a critical role of methylation in the process of IGC establishment (Herman & Sultan, 2011).

The work by Zheng et al. (2013) demonstrated subtle changes in DNA methylation in rice in response to drought for six generations; it was found that only the drought-sensitive variety responded in a meaningful way, while changes in the resistant variety were negligible (Zheng et al., 2013). Similarly, Arabidopsis plants exposed to drought exhibited only subtle stochastic changes in DNA methylation that did not accumulate in consecutive generations of drought exposure (Ganguly et al., 2017).

Exposure to many other stresses such as salt, flood, heat, cold and UVC also led to changes in DNA methylation in the progeny; in all these cases, global genome hypermethylation was observed (Boyko et al., 2010). As we mentioned above, changes in methylation often persist for several generations after stress has been removed. In Arabidopsis plants exposed to salt, water or temperature stress, hypermethylation persisted to a second generation when plants were propagated under normal conditions (Boyko et al., 2010).

Global genome hypermethylation in the progeny of stressed Arabidopsis plants does not reflect changes in the individual loci. Promoters of *SUVH2*, *SUVH5* and *SUVH8* genes involved in the regulation of the chromatin structure, and the promoter of *ROS1*, responsible for demethylation activities, were hypermethylated, while the promoters of stress-responsive genes UVH3, ERF1, TUBG1 and RAP2.7 were hypomethylated (Bilichak et al., 2012). As in the case of *Polygonum persicaria* plants described above, exposure of seeds of the progeny of salt-stressed plants to 5-azaC, a chemical compound that modifies cytosines by preventing methylation, removes the positive IGC in the form of stress tolerance and prevents the inheritance of hypermethylation (Boyko et al., 2010).

Similar to the changes in methylation found in response to abiotic stresses, global genome hypermethylation was also observed in the progeny of TMV-infected tobacco plants; hypermethylation persisted in the second generation propagated in a normal environment (Boyko et al., 2007). Loci that were undergoing rearrangements were found to be hypomethylated, while loci that were stable were either normally methylated or hypermethylated. It can be hypothesized that such differential methylation controls

the rearrangements in the genome of stressed plants (Boyko et al., 2007).

## 3. Possible mechanisms involved in the regulation of TGCs

What are the mechanisms that control heritable changes in response to stress? How is the specificity of changes established and how are they propagated? To understand it, we first need to understand how genetic information is normally inherited in plant gametes. Plant gametes are established late in development. Meiocytes differentiate from somatic meristematic cells. They differentiate into microspores and megaspores, and after several cell divisions, they give rise to pollen and ovum. Pollen consists of generative cell (GC) and vegetative cell (VC), and they differ in gene expression and the presence of siRNAs. While VC is hypomethylated and has considerably higher levels of expression of various genes, including those giving rise to siRNAs, the GC is fairly hypermethylated, with poor gene expression and low level of siRNAs. siRNAs expressed in VC can cross to GC where they are involved in the suppression of transposon activity (Martinez & Kohler, 2017).

Several mechanisms may be involved, and research demonstrates the role of RdDM, ncRNAs, DNA methylation and demethylation processes and histone modifications. The accumulation of metabolites, proteins or certain coding and non-coding RNAs may also play a role in the establishment of IGC, as they may give an advantage to the developing embryo. While all the above-mentioned molecules may contribute to IGCs, for TGCs, the involvement of metabolites, proteins or transcripts is highly unlikely, unless there is a certain mechanism of amplification of such metabolites or proteins, which has not yet been ruled out. Accumulation of stress-induced molecules is likely to affect female gametes more than male gametes, simply due to the larger cytoplasmic content of the former. Indeed, it was shown that epigenetic memory of salt stress is primarily established through the female gametes, while in the male gametes, changes in the DNA methylation were erased by the activity of DNA glycosylases, demonstrating both that heritable events are controlled by methylation and that there is a specific mechanism to restrict transmission of these events through male gametes (Wibowo et al., 2016).

### 3.1. The role of epigenetic regulators

Epigenetics is the most plausible mechanism behind heritable changes in response to stress. DNA methylation is likely to play the most crucial role. In plants, DNA methylation occurs in various sequence contexts, including symmetrical methylation at CG and CHG sites and asymmetrical methylation at CHH sites. Control of DNA methylation in plants is complex, with symmetrical CpG and CpHpG and non-symmetrical CpHpH methylation established and maintained through multiple, partially redundant mechanisms. *De novo* symmetrical methylation is established by the DOMAINS REARRANGED METHYLTRANSFERASE 2 (DRM2) with the help of ncRNAs of the RdDM pathway, while maintained by the METHYLTRANSFERASE 1 (MET1) in the CpG context and CMT2/CMT3 proteins in the CpHpG context (Zhang et al., 2018). CMT3 is recruited to the repressive histone mark H3K9me2 (Du et al., 2015), and in turn, CMT3 binding to DNA can facilitate the recruitment of H3K9me2 (Du et al., 2015). In contrast, CpHpH methylation is established by DRM2 and maintained by DRM2 at short transposons in euchromatic regions and by CHROMOMETHYLASE 2 (CMT2) at large transposons in heterochromatic regions (Zemach et al., 2013). DRM2 uses 24-nt siRNAs to guide DNA methylation at euchromatic TEs (Law & Jacobsen, 2010; Matzke & Mosher, 2014), while Decreased DNA Methylation I (DDM1) mediates recruitment of CMT2 to pericentromeric H3K9me2 regions (Stroud et al., 2014). The functionality of the RdDM pathway is also partially dependent on Dicer-like (DCL) proteins, DCL2, DCL3 and DCL4 (Yang et al., 2016). It can thus be hypothesized that the RdDM pathway is responsible for heritable changes in phenotype.

Experiments in our laboratory and the work of the others partially confirmed this hypothesis. We found that *dcl2* and *dcl3* plants but not *dcl4* plants exposed to UVC were impaired in IGCs in transposon activation, changes in leaf size, differential changes in the histone marks and expression of several repair genes (Migicovsky & Kovalchuk, 2014). The more prominent role of DCL2 and DCL3 as compared to DCL4 in the establishment of IGC and TGC was also confirmed in the progeny of Arabidopsis exposed to heat (Migicovsky et al., 2014).

Rasmann et al. (2012) obtained similar results—they found the Arabidopsis *dcl2 dcl3 dcl4* triple mutant impaired in passing the memory of exposure to herbivory to the progeny (Rasmann et al., 2012). Somewhat different results were reported by Ito et al. (2011); they found that the heat-induced expression of *ONSEN* was higher in the *dcl3* plants compared with the wild-type plants and suggested that DCL3 may be partially restricting the accumulation of *ONSEN* in response to heat stress in somatic tissues (Ito et al., 2011). They found a higher rate of transposition of *ONSEN* and new reinsertions in the progeny of heat-stressed *dcl3* plants. Hence, despite the fact that the authors reported somewhat different results than the two above-mentioned studies, they still suggested the role of siRNA biogenesis in the regulation of heritable response to stress. Likewise, the potential role of RdDM was also suggested for the response to the elevated levels of $CO_2$; changes in the plant physiology and changes in DNA methylation in the progeny were dependent on the function of RdDM machinery, specifically CMT2 and CMT3 DNA methyltransferases (Panda et al., 2023). It should be noted that the siRNAs may not be absolutely required for intergenerational memory, as the changes in the DNA methylation in the stressed plants can occur through the RDR6-RdDM pathway (Nuthikattu et al., 2013) or through the activity of DNA glycosylases (Williams et al., 2022).

DDM1 can also play a role in the establishment of heritable response to stress as it regulates the recruitment of CMT2 to DNA (Stroud et al., 2014). DDM1 mutant has substantial loss of DNA methylation and activation of transcription of many resistance genes. Furci et al. (2019) have analysed the pathogen tolerance of epigenetic recombinant inbred lines (epiRILs) obtained by crosses of *ddm1* mutant and wild-type Arabidopsis plants; the progeny of the cross maintained hypomethylated status of many loci in the absence of *ddm1* mutation for sixteen generations (Furci et al., 2019). They found several epigenetic quantitative trait loci (epiQTLs) associated with the priming of defence-related genes rendering plants resistant to biotrophic downy mildew pathogen *Hyaloperonospora arabidopsidis* (Furci et al., 2019). They further propagated these plants to F9 and F10 generations and confirmed that the resistance to this pathogen was retained, although it was lost in ~2.5% (2 of 40 families), and in the remaining families, considerable variations in the resistance were observed (Furci et al., 2019).

The mechanism of IGC and TGC may involve several steps. First, on the level of somatic cells, stress response includes differential expression of mRNAs, ncRNAs and changes in DNA

methylation and histone modifications. If stress occurs early during development and influences the whole plant, gamete cells that would derive from the meristem will acquire and propagate the signal. If stress occurs when gametes are established, they may also be altered in response to stress. Even if meristem cells or gametes are not altered directly, these cells may acquire information about stress from all other somatic cells through the active functions of plasmodesmata and phloem that circulate a variety of molecules, including ncRNAs (Maizel et al., 2020; Yang et al., 2023). It is possible that changes in DNA methylation and histone modifications caused by the RdDM mechanism may already occur in meristem cells or early during gametogenesis. Second, changes that occur in meristem cells or in the developing gametes have to survive reprogramming, a mechanism that erases the epigenetic marks, such as changes in DNA methylation, histone modifications and degradation of mRNA in pollen (Borg et al., 2021). Male and female gametes likely do not contribute to the heritable memory in an equal manner. It was shown that female gametes accumulate greater amount of polymerase IV (PolIV)-dependent ncRNAs than male gametes (Mosher et al., 2009). It is proposed that heritable response to stress is mainly under maternal control (Pecinka & Mittelsten Scheid, 2012). Although the evidence is scarce, at least one report by Wibowo et al. (2016) demonstrates that enhanced tolerance to hyperosmotic stress in the progeny is passed through the female germline (Wibowo et al., 2016). One of the DNA glycosylases, DEMETER (DME), is known to be especially active during male gametogenesis and is suggested to play a critical role in the eraser of methylation marks during the reprogramming step (Khouider et al., 2021). The authors exposed *dme-6* plants to hyperosmotic stress for two generations and found the progeny of these plants to be more tolerant to hyperosmotic stress as compared to the progeny of wild-type plants, suggesting that DME actively resetting the memory of stress in the male gametes (Wibowo et al., 2016). Also, much higher genome instability was observed in the progeny of UVC- and salt-stressed plants when the non-exposed pollen was used to pollinate the exposed ova, as compared to fertilization of the non-exposed ova with the exposed pollen (Boyko & Kovalchuk, 2010). It was also demonstrated in Arabidopsis that transgenerational phenotype aggravation in the Chromatin assembly factor-1 (CAF-1) mutant, impaired in chromatin assembly, was predominantly propagated by female gametes (Mozgova et al., 2018).

Epigenetic changes caused by stress also need to survive the second level of reprogramming that occurs after the fertilization event. It is possible that changes in DNA methylation occur in mature gametes or early embryos and are caused by differential expression of ncRNAs produced in gametes or embryos, or even in the endosperm. Third, it is possible that some of the differentially expressed ncRNAs may survive all reprogramming steps and trigger changes directly in the progeny. Our recent work in *Brassica rapa* showed that heat stress induces changes in ncRNA and mRNA expression in meristem tissues and gametes; some of these changes were propagated into the developing embryo and even into the progeny (Byeon et al., 2019).

Fourth, the propagation of stress memory and the maintenance of phenotypic changes in the next generations may require continuous stress exposure (generation after generation). This is not surprising because if changes in DNA methylation and ncRNA expression that trigger it play an essential role, they need to be generated constantly to reinforce transgenerational memory and replenish the molecules depleted during reprogramming.

It is curious that DNA methylation changes represent the most common TGC in the papers we described above. We can assume

that TGCs are triggered by differential expression of non-coding RNAs that target various genomic loci to establish differential methylation and differential gene expression, leading to changes in stress tolerance. DNA methylation is maintained more consistently regardless of whether plants are exposed to stress for the second time, while stress tolerance depends on the second stress exposure, which suggests that changes in DNA methylation are more robust and can persist in the absence of stress re-exposure.

## 4. Evolutionary significance of IGC and TGC, cost and benefits and maladaptation to stress

In this review, we have presented multiple examples of IGC and TGC in response to stress in plants and discussed the type of changes that occur and the potential mechanisms of their establishment. Are the IGCs or TGCs just examples of reprogramming escapes? Or is there a reason plants allow information about stress to be passed to the progeny?

When plants mount the defence against stress, they allocate resources from their growth and development programme to the response to stress. In this respect, the response to mild stress in the form of priming was developed as a mechanism to optimize the trade-offs of cost and benefit of higher tolerance to stress (Lopez Sanchez et al., 2021). The stressor may never appear again, and in this case, those plants that did not prime their defences have an advantage, as they have focused on growth and development instead of allocating resources to priming (Wilkinson et al., 2019). In contrast, those plants that mount priming will always be better off if stress is repeated during their growth or in the progeny. At the population level, some plants may receive more severe stress or be more genetically or epigenetically 'primed' to respond to stress with heritable change. It is even possible that there is a heterogeneous response within the same plant, where the level of response is gradual among all produced seeds. It would be interesting in the future to test this theory, focusing on the potential for the distance of dispersion of seeds to correlate with the degree of transgenerational response—the rate of changes may be proportional to the distance at which the seeds would land from their mothers.

The molecular mechanisms of somatic and transgenerational response have likely been established through thousands of generations of trial and error. There were likely cases when the cost of establishment of priming paid off because the stress repeated itself, and those plants that utilized it survived better and passed the genetic or epigenetic regulatory mechanisms to the progeny. Many theoretical papers were published attempting to correlate the response in the form of maternal effects (IGC or TGC) and changes in phenotypic plasticity with stress severity or intensity. It is proposed that maternal effects correlate with a periodicity of stress exposure. In a stable environment, maternal effects may have a slight negative influence on phenotypic plasticity, while in an abruptly changing environment that is maintained at a more or less constant level, maternal effects would have a strong positive influence allowing the progeny to adopt beneficial maternal phenotypes (Kuijper & Hoyle, 2015). In contrast, when there are fluctuations in the presence or severity of a stressor, maternal effects fluctuate or autocorrect according to the presence of a stressor (Figure 2a).

Generally, the strongest TGCs and maternal effects occur for those traits that are under very strong selective pressure, while for the traits that are under weak selective pressure, the evolutionary scope of maternal effects is very low or limited (Figure 2b). As

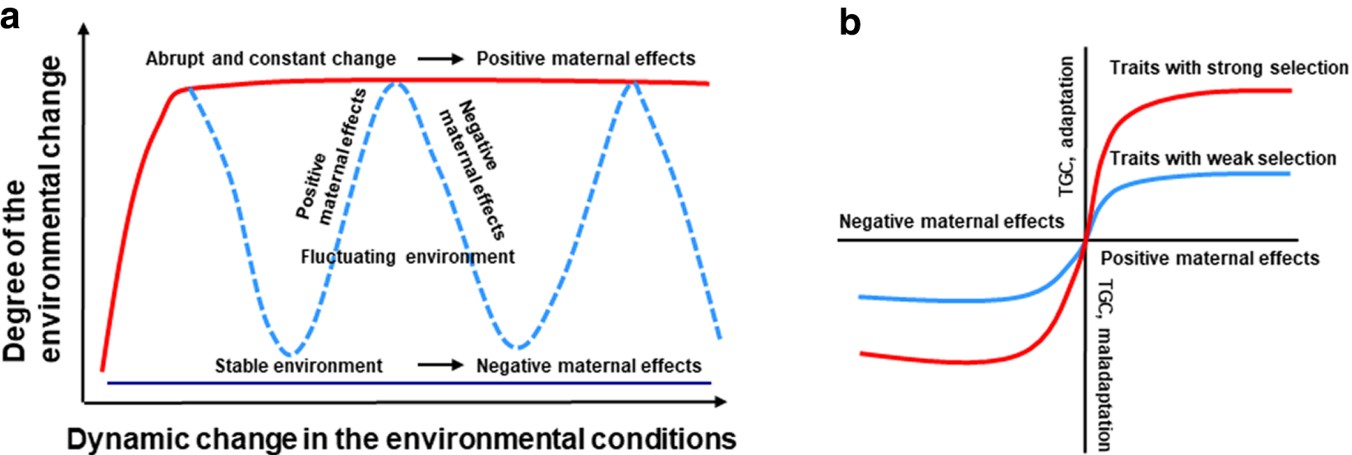

**Figure 2.** Correlation between environmental changes, maternal effects and adaptation. (a) Environmental stability regulates maternal effects. When environmental conditions are stable, maternal effects have a slight negative influence on selection or evolution and TGC. When environmental changes are rapid and stable, there is a positive maternal effect on TGC and trait diversity. Finally, when the environment fluctuates from stressful to normal, there is an equilibrium in maternal effects, changing from negative to positive. (b) TGC and adaptation or maladaptation are different for different traits. Traits under weak selection tend to respond less effectively to maternal effects and demonstrate lower TGC. In contrast, traits under higher selection pressure respond strongly to maternal changes; thus, the TGC and adaptation or maladaptation are easier to observe.

it appears, the vast majority of traits are under weak selection; therefore, it is more problematic to observe transgenerational phenomena in nature; in contrast, it may be easier to establish IGC or TGC in the laboratory, if you identify the trait under strong selective pressure (Kuijper & Hoyle, 2015).

At the end of the day, since priming as a response to stress has been demonstrated for many species, we assume that this mechanism is adaptive in nature. However, is transgenerational priming truly adaptive? We presented many examples where the progeny of primed plants had higher tolerance to the same stress and sometimes to a different stress. Very few reports, however, studied whether the fitness of such plants is comparable to the fitness of naïve plants when there is no encounter of stress in the progeny. Moreover, often great resistance to the stress encountered by parents results in lower resistance to another type of stress, and this is especially true for biotic stress encounters. There are several reports demonstrating the evidence of transgenerational maladaptation.

Repeated exposure to ozone-sensitized grapevine made them more sensitive in the progeny (Soja et al., 1997). Differential response to drought was found among closely related species, *Polygonum persicaria* and *Polygonum hydropiper*; while the progeny of the former one were more fit as compared to the control, the progeny of the latter one exhibited maladaptive traits—smaller seedlings with slower-growing roots (Sultan et al., 2009). The progeny of Arabidopsis plants exposed to spider mites were more resistant to infection with spider mites and even aphids but developed higher sensitivity to the biotrophic bacteria *Pseudomonas syringae* (Singh et al., 2017). The progeny of Arabidopsis plants exposed to the biotrophic pathogen *P. syringae* was more tolerant to infection with the biotrophic pathogen, *Hyaloperonospora arabidopsidis*, while being more sensitive to the necrotrophic fungus *Alternaria brassicicola* (Luna et al., 2012). Likewise, the progeny of Arabidopsis plants infected with biotrophic pathogen *P. syringae* or necrotrophic pathogen *Plectosphaerella cucumerina* or exposed to high salinity were more tolerant to the same pathogen but were more sensitive to a different pathogen—the progeny of plants exposed to the biotrophic pathogen were more sensitive to necrotrophic pathogen and vice versa; curiously, the progeny of salt-stressed plants did not acquire higher salt tolerance but

was slightly more tolerant to both pathogens (Lopez Sanchez et al., 2021). Another potential problem is that invasive species may have greater benefits from transgenerational plasticity, as it allows them to retain fitness in nutrient-rich environments and outperform other species in nutrient-poor environments; this was demonstrated for two invasive species, *Cyperus esculentus* and *Aegilops triuncialis* (Dyer et al., 2010).

## 5. Engineering plants with heritable epigenetic modifications

The knowledge we obtain from all inter- or transgenerational studies will allow us to understand how the memory of stress is formed and passed to the progeny. Information about loci that undergo epigenetic changes would allow us to engineer plants with higher stress tolerance.

Targeted epigenetic changes in the form of changes in DNA methylation and chromatin structure, leading to activation of multiple genes, have been demonstrated in plants; the dCas9-SunTag system fused to the VP64 transcriptional activator was used to target multiple loci for DNA demethylation; activation of FWA locus remained heritable for several generations (Papikian et al., 2019). More recently, Wang et al. used Clustered Regularly Interspaced Short Palindromic Repeats (CRISPR) or dCas9 in combination with the *TEN-ELEVEN TRANSLOCATION1* (*TET1*) demethylation domain (Wang et al., 2022) to target a naturally occurring hypermethylation epiallele (He et al., 2018) in one of the Arabidopsis ecotypes. They were able to achieve hypomethylation of the *PPH* gene, resulting in accelerated leaf senescence, inherited for two generations (Wang et al., 2022). Tang et al. also used the SunTag-dCas9-TET1cd system to target the *FIE1* gene (Tang et al., 2022); they found that the dwarf phenotype associated with hypomethylation of the *FIE1* gene was inherited for two generations. In the cases described above, the targeted locus was a locus with naturally occurring variations in methylation status. It remains to be shown whether targeted heritable DNA methylation changes can also be achieved in the other loci. Inheritance of DNA methylation pattern and associated phenotypes have also been demonstrated in mice; two metabolism-related genes, the ankyrin repeat domain 26 and

the low-density lipoprotein receptor, were targeted in embryonic stem cells, and the progeny with heritable obese phenotype was obtained (Takahashi et al., 2023).

## 6. Concluding remarks

In this review, we discussed the hypothesis that TGCs are caused by the differential expression of ncRNAs and RdDM mechanisms causing differential changes in DNA methylation and possibly histone modifications that escape reprogramming and give advantage to the progeny of stressed plants. Direct links between differentially expressed siRNAs causing changes in DNA methylation at specific loci and changes in stress tolerance remain to be established. It is unclear whether such siRNAs are passed from the progeny via gametes, or their expression is induced in the early developing embryo or the germinated plants by some other unknown mechanisms. It is also possible that such siRNAs are propagated in the cytoplasm through some amplification mechanisms, or by avoiding degradation, rather than through the activation of transcription. It remains to be shown whether differentially regulated siRNAs are stress-specific, are indeed directed towards specific loci in the genome and promote specific changes at epigenetic levels.

What is known, however, is that the expression of some of the ncRNAs and their fragments is heritable. It is also known that changes in the methylation pattern in the progeny occur at various hotspots, relevant to the encountered stress; in addition, the repetitive elements are hypermethylated to stabilize the genome, while many loci associated with stress tolerance are hypomethylated, likely to allow them to respond to stress more efficiently. It was documented that in most cases, changes in methylation in the progeny of stressed plants occur at CHH, suggesting the role of RdDM, as de novo methylation in this context and the maintenance of this methylation are assisted by RdDM. The potential role of RdDM was further supported by showing that several mutants impaired in RdDM were impaired in heritable changes in response to stress.

Despite the huge amount of work done, there are still a number of questions remaining.

It is still unclear whether heritable events, especially TGC, represent a true plant adaptive mechanism, or they are just 'imperfections', that escape from reprogramming. If RdDM and siRNAs are involved in the establishment of transgenerational events, why do we see so few reports implementing specific siRNAs in changes in methylation and phenotype? Also, why the changes in DNA methylation are frequently very massive, but the changes in phenotype are very subtle? All these questions remain to be answered by well-planned and carefully executed experiments.

Finally, we would like to apologize to all the scientists whose work we were not able to cite in the review.

## Acknowledgements

The author thanks the reviewers for making this review better.

**Financial support.** This study was funded by the Natural Sciences and Engineering Research Council of Canada Discovery Grant RGPIN-2023-03267.

**Competing interest.** The author declares none.

**Data availability statement.** There are no data to share.

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
