## [Reviewer Report]

It is my pleasure to submit this review that provides a comprehensive coverage of somatic and heritable changes in the response to stress. We discuss evolutionary meaning of heritable changes and provide many examples of heritable response to stress. We covered details of epigenetic response to stress. The review is not under consideration anywhere and I declare no conflict of interest.

Sincerely,

Igor Kovalchuk

---

## [Reviewer Report]

The author reviews “heritable responses to stress in the plants”. The abstract and introduction clarify that the type of “heritable responses” the review analyzes are what the literature refers to as “IGC” and “TGC”. A hallmark of IGC and TGC are that they are transient and reversible responses that occur in a more less systematic fashion among plants in a population, rather than being random events affecting individual plants. In that sense, IGC and TGC are evolved acclimation responses to stress. An good example is “priming”, where exposure to stressor in one generation can confer more effective stress responses in the direct offspring generation and sometimes even in generations following the offspring generation.

I have several major comments about this review.

- First, the review does not deliver on the topic that is laid out in the abstract and the introduction. It spends a lot of time talking about the impact of stress on genome stability and the rate of HR, which are non-systematic events that lead to non-reversible genomic change, can can therefore not explain the reversible nature of IGC and TGC, nor the fact that the IGC and TGC are are systematically induced in population under stress.

- Second, I don’t understand the rationale for the structure of the review. It starts with possible molecular mechanisms, then moves to a description of the phenotypic examples of IGC and TGC and then returns to molecular considerations. In my view, the review could benefit tremendously from providing a clearer structure for the reader.

- Third, there are numerous reviews on IGC and TGC. The author needs to lay out in the beginning what makes his review unique. What unique aspect is he tackling? How is this different from existing reviews?

- Forth, the authors often confuses stable epigenetic changes that have been reported with transient ones.

In sum, in my assessment, the review requires a clearer focus, a more logical structure and a clear demarcation from existing reviews on that topic. The review needs to decide if it is about general, heritable, molecular changes in response to stress (which would include changes in mutation rates, HR etc.), or if it is about IGC and TGC as they are used in the literature.

Additionally:

- The author writes: “However, it should be noted that no experiment comparing the efficiency of response to environmental stimuli of genetic variants vs epigenetic variants was ever conducted, thus our assumption is strictly a hypothesis.” What type of experiment would that be?

- The author writes: “Such changes in DNA methylation and chromatin structure represent

epimutations and could lead to the development of epialleles if stabilized.” What does the author mean by “stabilize”, mechanistically?

- The section “2.2 IGCs/TGCs in the form of changes in plant physiology and stress response” should come first. Here the author establishes the phenotypic relevance / evidence of IGC and TGC. Considerations regarding molecular mechanisms should come later. Also, the second heading is a bit of a misnomer, as discussion is mainly about stress phenotypes or types of stressors rather than about physiology.

- The author writes “Furthermore, in this review, we will not cover such transgenerational events as gene silencing..”, but spends time describing experiments that show exactly that (e.g. the work on epiRILs).

- The whole section “2.3 IGCs/TGCs in the form of changes in the genome stability” refers to molecular consequences of stress that cannot explain the reversibility of IGC and TGC at the phenotypic level.

- What does the author mean by “inheritance of recombination events”?

- The author writes “Genome destabilization in the progeny is likely not liner”. “Liner”?

- The review could benefit from some English editing. The main concern here is the omission of “the” in many places.

---

## [Reviewer Report]

The manuscript “Heritable responses to stress in plants” by Kovalchuk reviews the recent literature about stress responses in plants that are inherited across generations, and this inheritance is often mediated by epigenetic mechanism. I think the topic of this review is interesting, that being said there are a reviews that have covered this topic in recent years. Moreover, from a more broad perspective epigenetic inheritance has been reviewed extensively over the past years.

This manuscript does a fair job of reviewing the recent literature specific to plants. However, there is a certain lack of focus that I would like to see corrected. I think currently the manuscript groups some phenomena that are separate and it would benefit either having clear distinctions these phenomena or dropping some of them altogether.

In section 2. the manuscript discusses epialleles that have been observed to occur in nature. However, these presumably spontaneously occurring epialleles (as in van der Graaf et al. 2015) are a different phenomenon from phenotypic plasticity that is inherited across generations (as in Wibowo et al. 2016). Currently these phenomena are not clearly enough separeted in the manuscript. But even inherited plastic responses are based on an underlying genetic program (utilizing the plant RdDM pathway for example as a response to specific stress) while some of these epialleles can be spontenous changes independent of DNA sequence.

The third phenomena that is discussed in section 2.3 is stress induced increases in mutation and recombination rates. Some research suggest that this increase in these rates is propagated into following generations even in the absence of the original stress. Again, this phenomena is quite different from phenotypic plasticity from an evolutionary perspective. Of course increased mutation or recombination rates can be beneficial for adaptation to a new environment.

I would suggest dropping the section about epialleles from this review and focusing on plastic responses and making a clear distinction between phenotypic plasticity and increases in mutation and recombination rates, as they have different evolutionary consequences.

I also think that section 4 does not review what is known about the evolutionary significance of across generation phenotypic plasticity very thoroughly. There has been a lot of theoretical work done to investigate in which conditions such plasticity should evolve and what are its evolutionary consequences, but none of this work is cited. Please consider citing some of this work. For example: the work of Bram Kuijper and others.

Kuijper, B. & Hoyle, R. B. 2015. When to rely on maternal effects and when on phenotypic plasticity? Evolution 69:, 950-968

There are also other theoretical papers that have investigated these issues.

I would also consider the evolutionary consequences of increased mutation and recombination rates separately, as they have different consequences on adaptation than phenotypic plasticity. Of course conditionally increased mutation and recombination rates can be adaptive.

Minor points

In the future, please include line numbers in the manuscript. This makes making comments easier.

The text has some points where terminology is needlessly complicated. The text speaks about immediate progeny (when discussing intergenerational inheritance) and then in the case of transgenerational inheritance speaks about the next generation (next generation of immediate progeny). It would be better to call them just progeny and grand progeny to distinguish between intergenerational and transgenerational.

Page 13, somewhat repetitive, these examples were discussed before and now are again the the perspective of the progeny.

Page 19, I think there is more evidence that maternal inheritance is more common. Check work from the lab of Sonia Sultan.

Herman, J. J.; Spencer, H. G.; Donohue, K. & Sultan, S. E. 2014. How stable ‘should’ epigenetic modifications be? Insights from adaptive plasticity and bet-hedging. Evolution 68: 632-643

Page 14, paragraph beginning “An interesting paper...” Wibowo is cited twice at the end of first sentence.

Same problem in page 22, when citing Elsalahy et al. 2020.

References

van der Graaf, A.; Wardenaar, R.; Neumann, D. A.; Taudt, A.; Shaw, R. G.; Jansen, R. C.; Schmitz, R. J.; Colomé-Tatché, M. & Johannes, F. 2015. Rate, spectrum, and evolutionary dynamics of spontaneous epimutations. Proceedings of the National Academy of Sciences, 112: 6676-6681

Wibowo, A.; Becker, C.; Marconi, G.; Durr, J.; Price, J.; Hagmann, J.; Papareddy, R.; Putra, H.; Kageyama, J.; Becker, J.; Weigel, D. & Gutierrez-Marcos, J. 2016. Hyperosmotic stress memory in Arabidopsis is mediated by distinct epigenetically labile sites in the genome and is restricted in the male germline by DNA glycosylase activity. eLife 5: e13546

---

## [Reviewer Report]

Dear authors,

Both reviewers and myself acknowledge that the topic of your review is interesting and timely. However, the reviewers emit strong criticisms on the structure and content of the manuscript. Both reviewers suggest to have different focus and ideas for re-organizing (removing some parts), in order to make the review unique (and different from the large amount of published reviews on this topic). Please do address in a revised version fewer topics than intended which can then be covered in greater depth for the benefit of the readers. The reviewers also suggest to carefully edit the English writing and especially to be more precise in your definitions.

Best regards,

Aurelien Tellier

---

## [Reviewer Report]

The manuscript has been revised and the structure is now more clear. Although I disagree with the choice to drop the genome instability part and keep the epiallele part, but fair enough.

The issue that there are a lot of recent reviews about this topic remains. While I have no scientific objections, I leave the decision whether the review is novel enough for the editor to decide.

My other points have been adressed in the revised manuscript.

Minor points (note that line numbers refer to the PDF proof.)

line 376: Is van der Graaf et al. 2015 cited correctly here? That paper is not about heritable responses to stress but abuot spontaneous epimutations.

Line 446 sentence should have the word “are”

---

## [Reviewer Report]

The author has substantially refocused and streamlined his manuscript based on the reviewer’s comments. I think this has improved the manuscript. Still, several conceptualizations / discussions throughout the work remain debatable (e.g. the distinction between stable epialleles and plasticity), and I occasionally do not share the author’s view on things. But the purpose of a review / opinion paper is to partly stimulate discussion, as it attempts to connect a diversity of empirical observations into a single perspective.

---

## [Reviewer Report]

Dear authors,

Both reviewers are now pleased by your revised version. It reads much better and is more focused.

One reviewer asked for some last minor corrections, please update your last submitted version accordingly.

Congratulations and thank you for your contribution.

Aurelien Tellier